# Characterisation of the T-cell response to Ebola virus glycoprotein amongst survivors of the 2013–16 West Africa epidemic

T. R. W. Tipton [1✉], Y. Hall[1], J. A. Bore[2], A. White [1], L. S. Sibley[1], C. Sarfas[1], Y. Yuki[3], M. Martin[3], S. Longet [1], J. Mellors[1], K. Ewer [4], S. Günther [5,6], M. Carrington [3,7], M. K. Kondé[2] & M. W. Carroll[1,8]

*Zaire ebolavirus* (EBOV) is a highly pathogenic filovirus which can result in Ebola virus disease (EVD); a serious medical condition that presents as flu like symptoms but then often leads to more serious or fatal outcomes. The 2013–16 West Africa epidemic saw an unparalleled number of cases. Here we show characterisation and identification of T cell epitopes in surviving patients from Guinea to the EBOV glycoprotein. We perform interferon gamma (IFNγ) ELISpot using a glycoprotein peptide library to identify T cell epitopes and determine the CD4+ or CD8+ T cell component response. Additionally, we generate data on the T cell phenotype and measure polyfunctional cytokine secretion by these antigen specific cells. We show candidate peptides able to elicit a T cell response in EBOV survivors and provide inferred human leukocyte antigen (HLA) allele restriction. This data informs on the long-term T cell response to Ebola virus disease and highlights potentially important immunodominant peptides.

[1] National Infection Service, Public Health England, Porton Down, Salisbury, UK. [2] Center for Training and Research on Priority Diseases including Malaria in Guinea (CEFORPAG), Nongo, Conakry, Guinea. [3] Basic Science Program, Frederick National Laboratory for Cancer Research in the Laboratory of Integrative Cancer Immunology, National Cancer Institute, Frederick, MD, USA. [4] The Jenner Institute, Oxford, UK. [5] Bernhard Nocht Institute for Tropical Medicine, Hamburg, DE, Germany. [6] German Center for Infection Research (DZIF), Partner Site Hamburg-Lübeck-Börstel-Riems, Hamburg, DE, Germany. [7] Ragon Institute of MGH, MIT, and Harvard, Cambridge, MA, USA. [8] Nuffield Department of Medicine, University of Oxford, Oxford, UK. ✉email: tom.tipton@phe.gov.uk

2013–16 saw the largest recorded epidemic of Ebola virus disease (EVD), resulting in over 30,000 cases and 11,000 case fatalities[1]. During this period efforts were made to establish new and experimental therapeutics, this has culminated in two lead candidate vaccines which have undergone extensive safety and efficacy trials[2,3]. The lead candidate vaccine uses Vesicular stomatitis Indiana virus (VSV) as a viral vector, and is known as rVSV-ZEBOV. This vaccine incorporates the glycoprotein (GP) of the Kikwit *Ebolavirus* from the 1995 epidemic on the capsid surface, in place of its native GP. This substitution has resulted in a loss of tropism of VSV for its target, however, subsequent clinical trials investigating the safety of rVSV-ZEBOV reported mild to moderate adverse events in a small number of participants[3–5]. There are a number of large animal studies demonstrating the efficacy of the rVSV-ZEBOV vaccine and a large clinical trial in Guinea has shown the vaccine to be effective and appropriate for ring vaccination strategy, with a reported 100% efficacy (95% CI 79·3–100·0; $p = 0.0033$)[6]. However, emerging evidence suggests that there have been cases of disease breakthrough associated with this vaccine[7].

Another vaccine candidate to be developed utilises a recombinant chimpanzee adenovirus subgroup 3 virus as a vector for EBOV, Mayinga strain, GP (ChAd3-EBO−Z). Research has investigated this vaccine on its own or in combination with a modified Vaccinia Ankara (MVA-BN-Filo) boost. The MVA-BN-Filo boost encodes for the same Mayinga strain GP as the ChAd3-EBO−Z, as well as the *Sudan ebolavirus* GP and Marburg virus GP, in addition, MVA-BN-Filo encodes for the Taï-Forest *Ebolavirus* nucleoprotein (NP)[2,8]. A close relation to this vaccine combination has been developed and has recently received marketing authorisation from the European Union medicines agency[9]. Both candidate vaccines are continuing to show success in the field, however, to what extent these vaccines need to mediate a cellular or humoral response to provide protection is still unclear.

Evidence from animal studies and survivor cohorts are helping us understand the naturally acquired immune response which in turn will help inform on vaccine design and may help elucidate the comparative need for a humoral or cellular response. Early work investigated the T cell response to mice vaccinated with Venezuelan equine encephalitis virus replicons, which expressed various EBOV proteins. This work found murine antigen-specific T cells to these EBOV proteins were generated, including the NP and GP. These T cells were expanded in vitro and adoptively transferred to EBOV naïve mice, when mice were challenged with an adapted EBOV strain it was found that they were protected from EVD[10]. Seminal evidence for the importance of T cells to EVD survival following vaccination comes from the work of Sullivan et al. who vaccinated non-human primates (NHPs) with human recombinant adenovirus serotype 5 (rAdHu5) which encoded for EBOV GP. Cynomolgus macaques were vaccinated then exposed to EBOV. Interestingly, if post vaccinated animals underwent T cell depletion using an anti-CD3 monoclonal antibody (mAb) they lost their ability to control disease and succumbed to infection. Furthermore, if prior to challenge primates were CD8+ T cell depleted using a monoclonal antibody then, again, they were unable to control disease, this was not the case for CD4+ T cell depletion prior to challenge[11]. However, work by Marzi et al. looking into the role of T cells following rVSV-ZEBOV vaccination in NHPs showed that CD8+ T cells were in fact dispensable and the humoral response, mediated by CD4+ T cells, was critical to vaccine-mediated protection[12].

Antibody and T cell responses have been shown to be long-lived amongst EVD survivors[13]. Therefore, the investigation into the natural immune response to EBOV may help better inform on vaccine design and the relative importance of cellular or humoral

immunity. Recent work found that during the 2013–16 West Africa epidemic patients with elevated levels of the T cell inhibitory molecules PD-1 and CTLA-4 were more likely to succumb to disease[14] and longitudinally characterised T cell response in two western repatriated patients, found a decrease in CD4+ T cells leading to a flip in the CD4+:CD8+ T cell ratio. It was also found that T cells showed elevated PD-1 expression and that there was impaired IFNγ production which was associated with virus reactivation[15]. Similarly, work by McElroy et al. investigated the cellular response to four acute EBOV infected patients at Emory University hospital, where they found striking activation of both CD4+ and CD8+ T cells to several EBOV proteins[16]. Work by Sakabe et al. 2018 identified a number of antigen-specific T cells amongst survivors of the 2013–16 West African epidemic and concluded that responses to the NP were immunodominant—suggesting NP should be included in any vaccine design[17]. Here, we assess the T cell response from a number of EVD survivors to the EBOV glycoprotein, we inform on a number of antigen-specific peptides, the resulting T cell phenotype and the associated HLA dynamics of the cohort studied.

## Results

IFNγ is a potent antiviral cytokine which is critical to the control and elimination of many intracellular pathogens. It is primarily produced by natural killer cells and antigen specific CD4+ and/or CD8+ T cells[18]. To determine to what extent survivors (two years post recovery) of EVD can mount a long-term immune response to the EBOV GP we used ELISpot to measure IFNγ release following overnight peripheral blood mononuclear cell (PBMC) stimulation with a GP peptide library. This peptide library consists of 187 peptides, each 15 amino acids long, overlapping by 11 thereby offset by 4 amino acids (Supplementary Fig. 1)[2]. Comparison of the summed frequency of IFNγ spot forming units (SFU) measured in response to stimulation with the peptide library indicated that EVD survivors have significantly elevated GP-specific IFNγ SFU frequencies compared with negative controls ($p < 0.0001$) with a median value of 331 SFU amongst the survivors and 6 amongst the negative controls. It can also be seen in Fig. 1c that the majority of EVD survivors are mounting a T cell response to the soluble region of the GP and that these T cell responses correlate with whole virus antibody levels (Supplemental Fig. 2). Furthermore, individual EVD survivor response to peptide pools (Fig. 1c and Supplementary Fig. 3) showed considerable heterogeneity, although the majority of survivors responded to peptides within the GP1-2 and GP1-4 peptide pools.

We next performed more in-depth peptide screening, via IFNγ ELISpot, to each individual peptide within our glycoprotein peptide pool. This analysis was performed on 15 EVD survivors, fresh in the field (Fig. 2a, Supplementary Fig. 4) and highlights a number of immunogenic regions, particularly within GP1-2 and GP1-4. Additionally, frozen PBMC that were transported back to the UK were used to perform peptide mapping on the EVD survivors who showed a response to peptide pools SP, GP1-1, GP1-2 or GP1-4 (Fig. 2b–e). Frozen samples were chosen for more in-depth mapping if they had an ELISpot reading that was five standard deviations above the mean negative value for the corresponding peptide pool shown in Fig. 1. From these additional IFNγELISpot studies we found several candidate peptides generated an IFNγ response, in particular, peptides 79 and 82 from peptide pool GP1-4 were studied in greater depth due to this region previously being highlighted as potentially immunogenic after vaccination[19].

To determine the contribution of either CD4+ or CD8+ T cells in the response seen to peptide pool GP1-4 we used survivor PBMC depleted for either CD4+ or CD8+ T cells and then

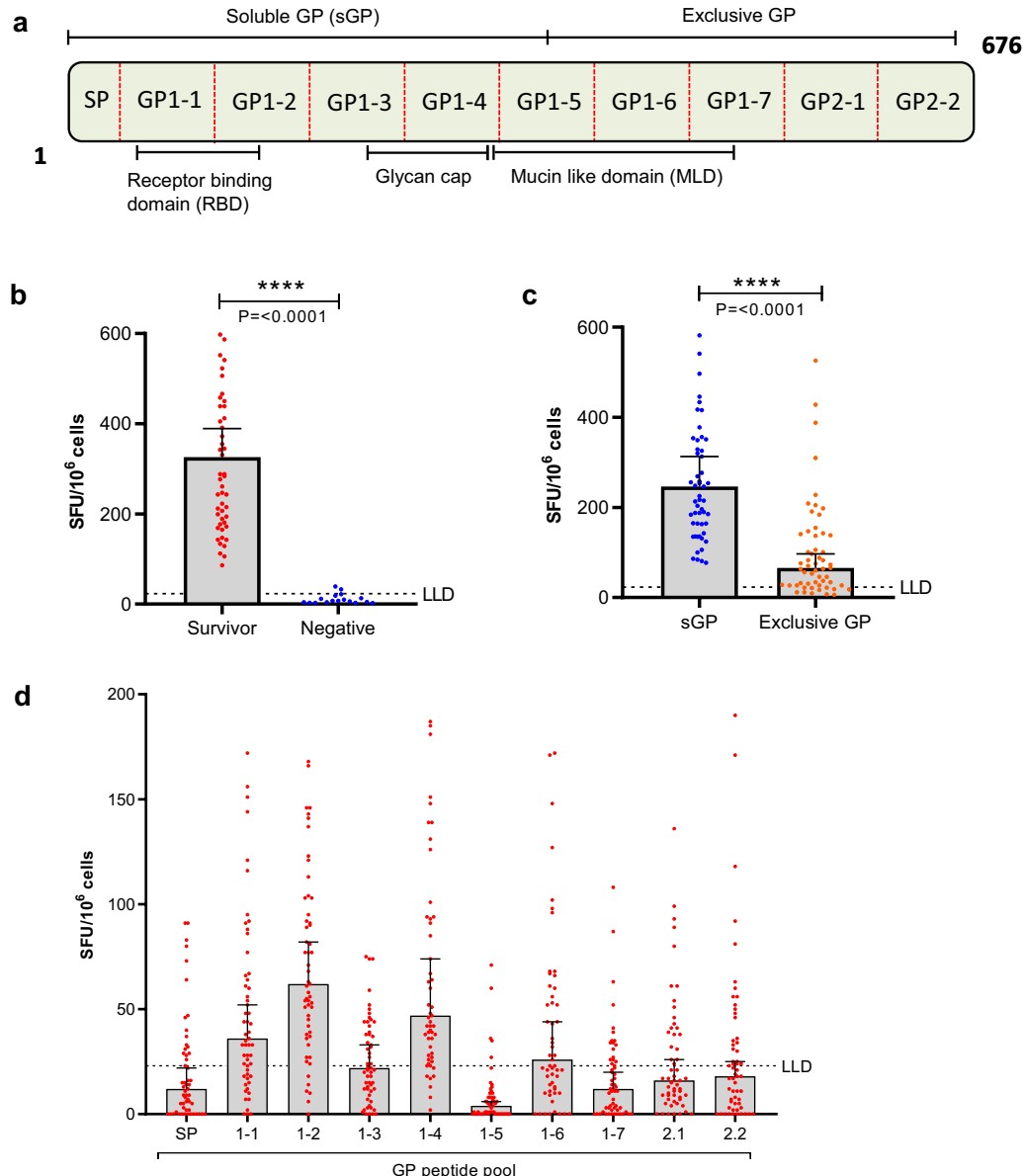

**Fig. 1 Cellular immune response, using fresh PBMC, to EBOV GP (Mayinga) peptide library as measured by IFNγ ELISpot. a** Schematic representation of the EBOV glycoprotein highlighting notable regions and peptide pools used in ELISpot analysis. SP = Signal peptide. **b** The summed ELISpot response to all GP peptides in the library amongst 57 EVD survivor and 18 non-exposed, negative, PBMC samples. **c** ELISpot response amongst 57 EVD survivors to either sGP or exclusive GP portions of the glycoprotein. **d** The ELISpot response amongst 57 EVD survivors to each peptide pool. For graphs **b-d** bars represent the median values with the upper 95% confidence interval. Two-tailed Mann–Whitney U test used to look for significance in (**b**) ($p$ = <0.0001) and two-tailed Wilcoxon test in (**c**) ($p$ = <0.0001). Dashed black line is the lower limit of detection (LLD) represents the in house cut off value (23 SFU), this is the mean of all negative results in (**b**) plus 3 standard deviations (SD) and discriminates between a positive and negative responder.

assayed for their IFNγ response to peptides that make up the GP1-4 peptide pool (Fig. 3). It was found that the peptides within sub-pool 2 were responsible for CD4[+] T cell activation ($p$ = 0.0264) whereas the peptides within sub-pool 3 were responsible for CD8[+] T cell activation ($p$ = 0.0255). GP1-4 sub-pool 2 contained peptides 74–80 and included peptide 79 which from Fig. 2a, e appeared to be immunogenic. GP1-4 sub-pool 3 contained peptides 81–88, which included peptide 82, again this looked to be immunogenic in Fig. 2a, e.

We next used flow cytometry studies to better characterise the immune response seen in Fig. 3. EVD survivor PBMC samples were stimulated overnight with GP peptide pool (all 187 peptides), peptide 79, peptide 82 or Staphylococcal enterotoxin B (SEB), which was used as a positive control. The following day,

cells were stained and acquired on the flow cytometer (Supplementary Fig. 5). The phenotype of cells that produced IFNγ and TNFα (double positive) or IFNγ and TNFα and IL-2 (Triple positive) in response to GP peptide pool stimulation can be seen in Fig. 4a, b. With regards to CD4[+] T cells this phenotype primarily expressed CD45RO[+] and CCR7[+/-] consistent with a central memory phenotype, in contrast, antigen-specific CD8[+] T cells, were CCR7[+/-] and CD45RO[-] which is indicative of a naïve or effector cell phenotype[20]. Antigen specific CD8[+] cells were also CD28[+], CD95[+] (Supplementary Fig. 6) and CD107a[+]. CD28 is a co-stimulatory receptor found on all T cells and its expression is elevated on antigen-experienced cells. CD95 is a death receptor and its expression is also increased on all memory subsets, whereas CD107a is a lysosome-associated membrane

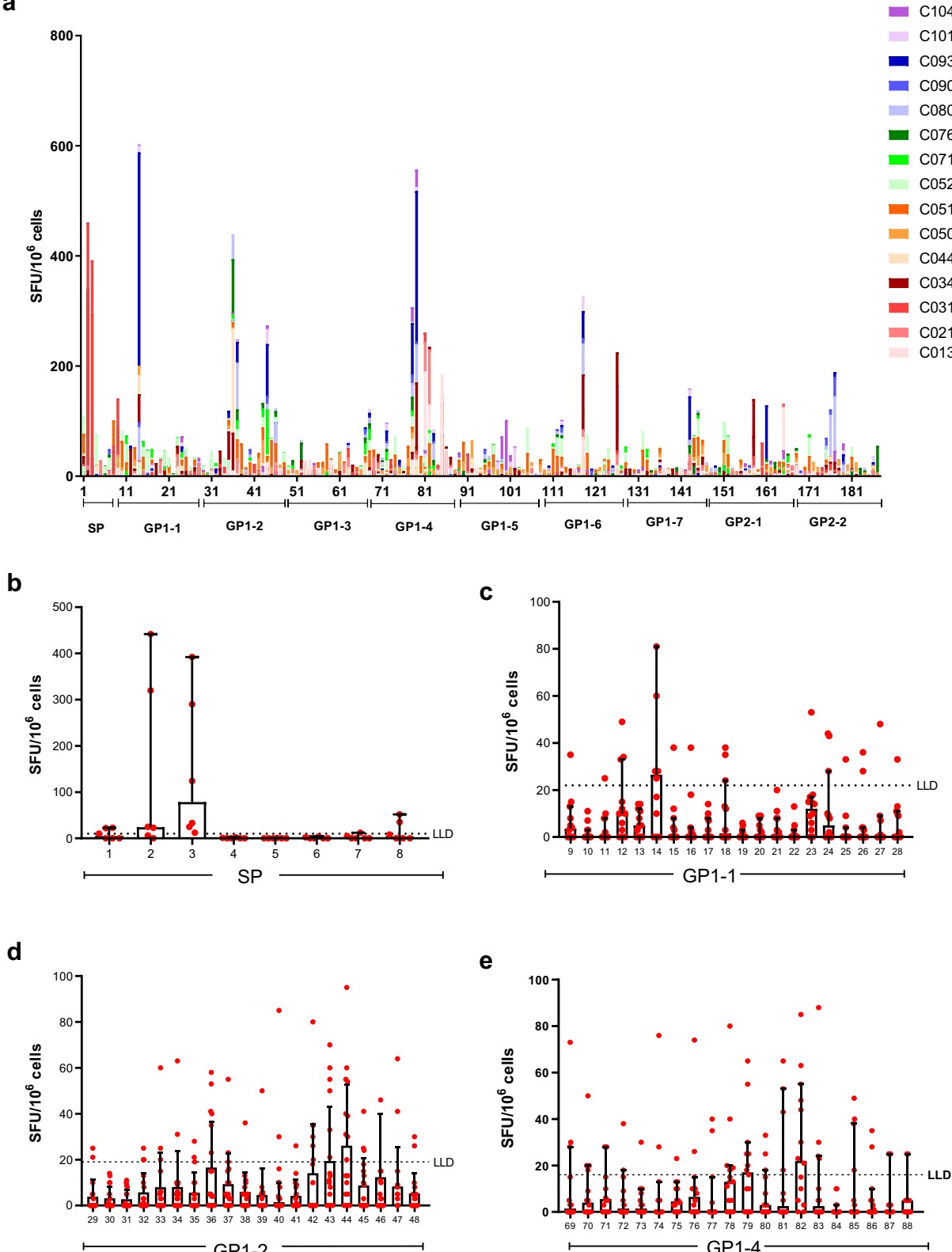

**Fig. 2 Cellular immune response to EBOV GP peptides as measured by IFNγ ELISpot. a** Response to 187 individual glycoprotein peptides amongst 15 fresh PBMC samples. Bars are stacked and indicate the sum of all results to each peptide amongst 15 EVD survivors. **b** Response amongst 6 EVD survivors to individual 15 mer peptides which make up the SP peptide pool. **c** The response amongst 12 EVD survivors to individual 15 mer peptides that make up the GP1-1 peptide pool. **d** Response amongst 18 EVD survivors to individual 15 mer peptides which make up the GP1-2 peptide pool. **e** The response amongst 14 EVD survivors to individual 15 mer peptides that make up the GP1-4 peptide pool. For graphs (**b**–**e**) red dots indicate individual data points and bars with error represent the median with the upper 95% confidence interval. Black dashed line represents the lower limit of detection (LLD) for the GP1-2 (19 SFU) or GP1-4 (16 SFU) peptide pool respectively, this was calculated using the mean of the negative samples plus three SD.

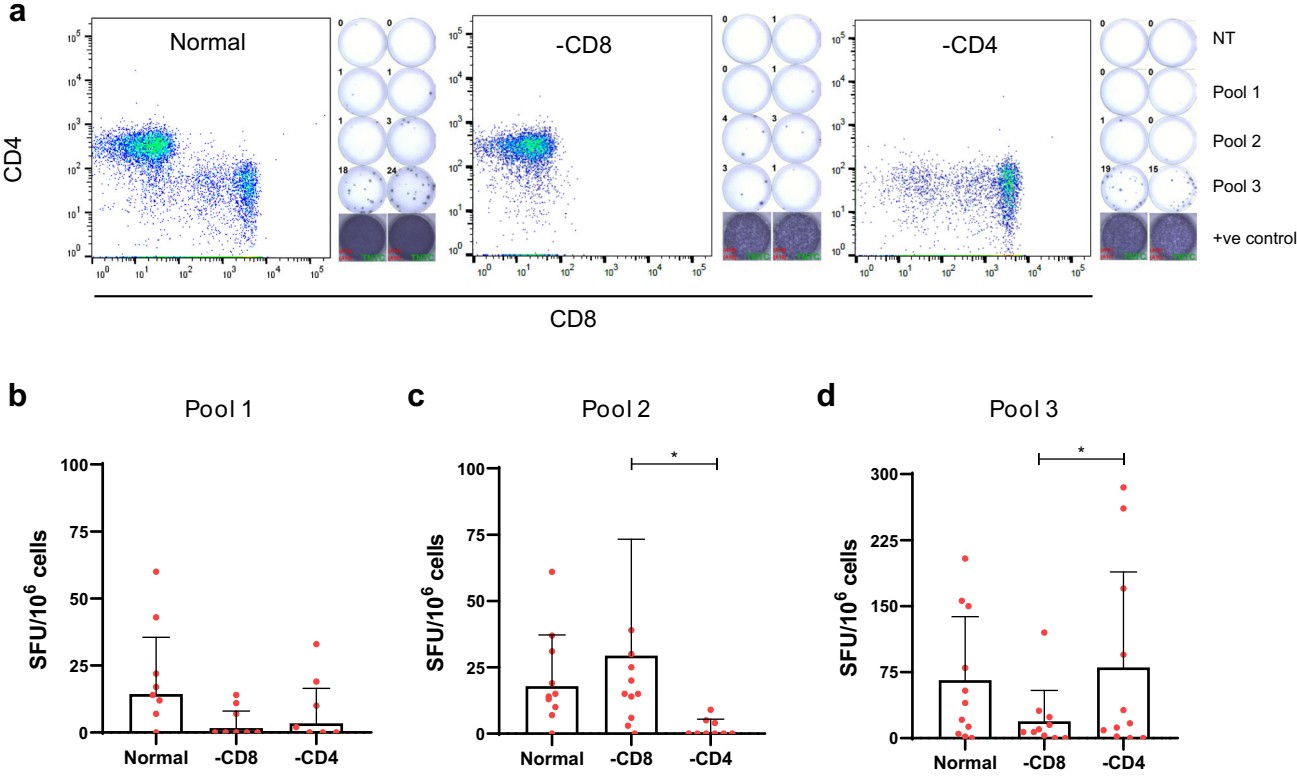

**Fig. 3 EVD survivor T cell memory response to EBOV glycoprotein GP1-4.** IFNγ ELISpot using EVD survivor PBMC that were depleted of either CD4+ (−CD4) or CD8+ (−CD8) T cells. **a** Representative flow cytometry and ELISpot images for survivor C052 showing that PBMC were successfully depleted for the desired T cell population and the corresponding ELISpot response to the various stimulation conditions. **b**–**d** Response to GP1-4 peptide sub-pools. (**b**) shows the response to sub-pool 1 which consists of peptides 69–73, (**c**) shows the response to sub-pool 2 which contains peptides 74–80 and (**d**) shows the response to sub-pool 3 which contains peptides 81–88. Data show the mean +SD of 11 EVD survivor samples. One-way ANOVA with repeated measures used for statistical analysis in (**c**) $p = 0.0264$ and in (**d**) $p = 0.0255$, $n = 11$ biological independent samples examined over 2 independent experiments.

glycoprotein and its expression has been associated with CD8+ T cell degranulation[20,21]. Antigen-specific CD4+ T cells showed a tendency to produce IFNγ, TNFα, and IL-2 whereas CD8+ antigen-specific cells primarily produced IFNγ and TNFα only, which is a functional profile consistent with the phenotypes described above (Fig. 4b). With regards to CD107a expression, there was a trend for CD8+ EVD survivor T cells to express more CD107a in response to GP peptide stimulation however this did not reach significance (Supplementary Fig. 7).

With regards to characterising the T cell response to individual peptides that were found to elicit an IFNγ ELISpot response we found that peptide 82 elicited a CD8+ specific T cell response (Fig. 4d) which was primarily associated with IFNγ and TNFα production, cytokine-producing cells again showed a CCR7+/− and CD45RO− phenotype. In contrast, peptide 79 was primarily associated with CD4+ T cells producing IFNγ, TNFα and IL-2 (Fig. 4c). Additionally, we found that peptide 3 was associated with a CD8+ T cell response, unfortunately, we could not determine whether the ELISpot response seen for peptides within GP1-2 were CD4+ or CD8+ mediated (Supplementary Fig. 8).

Finally, genomic DNA was used to HLA genotype EVD survivor samples and the most common MHCI and MHCII frequencies can be seen in Supplementary Fig. 9. We have shown that peptide 82 elicits a CD8+ T cell response, therefore, to determine the most likely candidate HLA alleles responsible for binding peptide 82 we performed in silico analysis using the immune epitope database and analysis resource (IEDB) research tool. Results in Table 1 indicate that as expected a number of

different HLA alleles are capable of presenting common fragments of peptide 82 to CD8+ T cells.

## Discussion

We were readily able to detect T cell responses to EBOV GP amongst survivors of EVD years after infection, however, to what extent these cytotoxic responses are important to acute infection is still being debated. Previous work by Dahlke et al. studied the T cell response amongst a repatriated EVD survivor in Germany. They found that CD8+ T cells dominated during the recovery phase of EVD and GP-specific CD8+ T cells were detectable but of low magnitude at 46 days post-recovery. However, this patient received multiple experimental treatments which may have altered lymphocyte dynamics during the recovery phase[22]. We again see GP-specific responses of low magnitude within our cohort which is on average greater than 2 years post EVD recovery. Comparison with vaccine data will be important and it has been shown that T cell responses to the same GP peptide pool are detectable 12 months post-rVSV-ZEBOV vaccination[23]. Recent work by Powlson et al. has characterised the T cell response to EBOV GP following vaccination with ChAd3-MVA and showed a number of immunogenic peptides, one of which (TTIGEWAFW) falls within the motif we found to elicit a CD8+ T cell response amongst EVD survivors (peptide 82)[19]. Based on our IEDB predictions we believe that this peptide is able to bind to a number of different HLA and therefore could provide a broad response amongst various ethnic backgrounds which

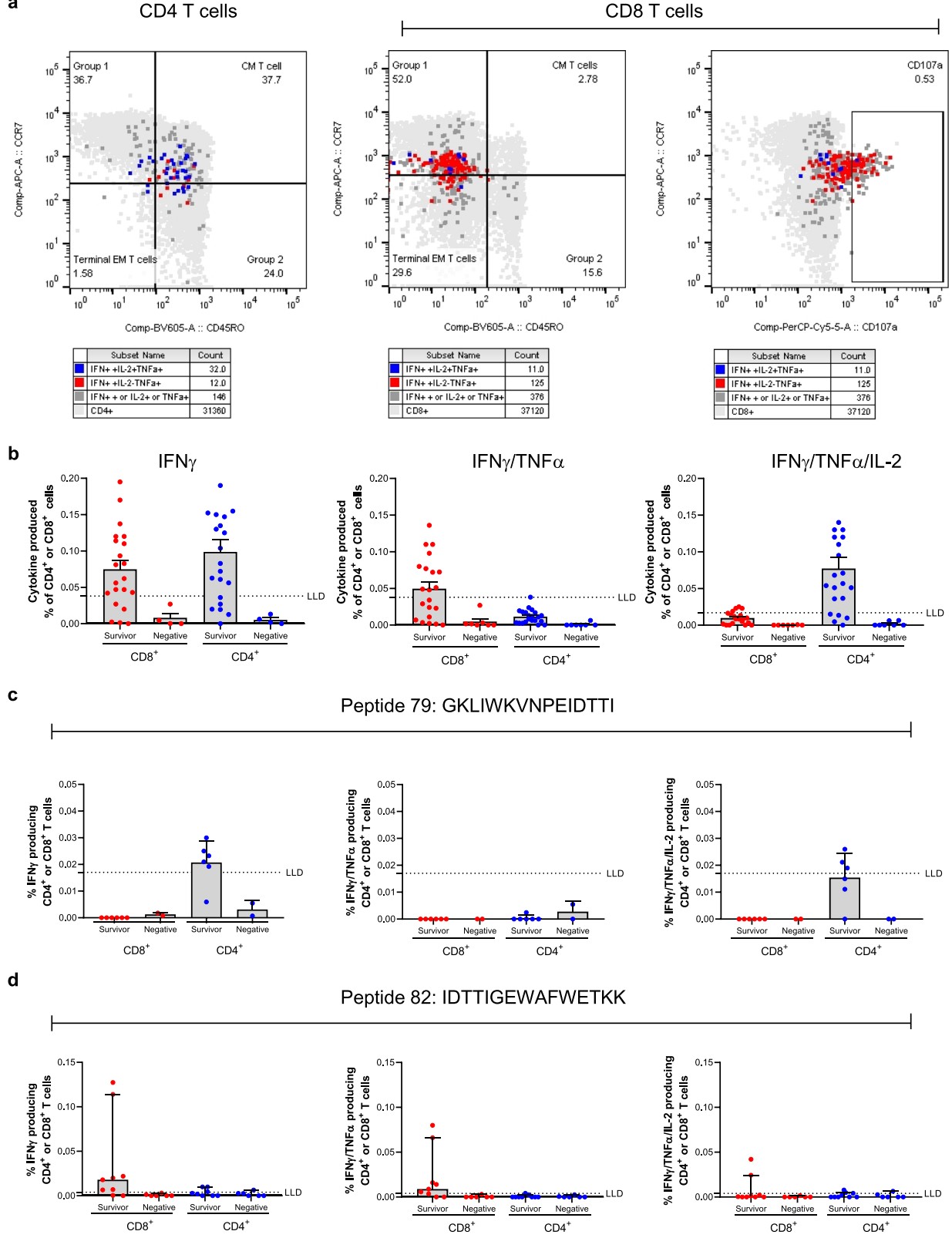

would favour its incorporation into future therapeutic vaccine platforms. Work by Ahmad et al. used immune-informatics tools to make in silico predictions on potential B and T cell epitopes of interest[24]. One of their top hits (IRGFPRCRY) was contained within our peptide 36, which we show here in Fig. 2 to give a response by IFNγ ELISpot, although we were unable to categorically determine whether this peptide resulted in CD4+ or

CD8+ T cell activation. Nevertheless, our results would support the validity of these reverse vaccinology approaches in vaccine design.

Important work by Sakabe et al. used an expression system to probe the immunogenicity of various EBOV proteins amongst EVD survivors from Sierra Leone and they found a broad response to various EBOV products. Interestingly, only half of the

**Fig. 4 Flow cytometry studies to characterise the EVD survivor T cell response.** 21 EVD survivor or 7 unexposed, negative, control PBMC were either left untreated or stimulated with EBOV GP peptides overnight. **a** Phenotype of Antigen Specific T cells following stimulation with *EBOV* GP peptide pool (187 peptides), representative data from Survivor C070. Cytokine combinations have been overlaid on top of the T cell phenotype. **b** The cytokine combinations from 21 EVD survivors, produced by either CD8+ or CD4+ T cells following EBOV glycoprotein peptide pool stimulation. **c** 6 EVD survivor and 4 unexposed, negative, control PBMC were either left untreated or stimulated with EBOV GP peptide 79 overnight. **d** 9 EVD survivor and 6 unexposed, negative, control PBMC were either left untreated or stimulated with EBOV GP peptide 82 overnight. Samples were analysed using FlowJo v10 and GraphPad v8. For graphs (**b**–**d**) red or blue dots indicate the individual data points and grey bars represent the median value with the upper 95% CI. LLD is specific to the respective cytokine combination and was determined by taking the mean +3 SD of the negative group.

**Table 1 In silico predictions for peptide 82/HLA binding to CD8+ TCR.**

| ID | Sequence | HLA-A_1 | HLA-A_2 | HLA-B_1 | HLA-B_2 | HLA-C_1 | HLA-C_2 | ELISpot (SFU/$10^6$ cells) | ICS (% IFNγ+/TNF+) |
|---|---|---|---|---|---|---|---|---|---|
| C013 | I**DTTIGEWAF**WETKK | 02:01 | 68:02 | **35:01** | 53:01 | 04:01 | 16:01 | 130 | 0.066 |
| C021 | IDTTIGEWAFWETKK | 30:01 | 68:02 | 18:01 | 42:01 | 05:01 | 17:01 | 99 | 0.000 |
| C105 | ID**TTIGEWAFW**ETKK | 02:01 | **26:01** | 45:01 | 56:01 | 01:02 | 16:01 | 85 | 0.080 |
|  | IDTTI**GEWAFWET**KK | 02:01 | 26:01 | **45:01** | 56:01 | 01:02 | 16:01 |  |  |
| C003 | IDTTIGEWAFWETKK | 02:01 | 23:01 | 51:01 | 53:01 | 06:02 | 16:01 | 63 | 0.009 |
| C126 | ID**TTIGEWAFW**ETKK | 02:05 | 33:01 | 52:01 | **57:03** | 04:01 | 18 | 55 | 0.006 |
| C092 | IDTTIGEWAFWETKK | 24:02 | 33:01 | 27:05 | 40:02 | 02:02 | 02:02 | 48 | 0.016 |
| C078 | I**DTTIGEWAF**WETKK | 03:01 | 30:01 | **35:01** | **35:01** | 04:01 | 16:01 | 44 | 0.000 |
| C081 | ID**TTIGEWAFW**ETKK | 23:01 | 30:02 | 07:02 | **58:01** | 04:01 | 07:01 | 24 | 0.014 |
| C093 | IDTTIG**EWAFWETKK** | 02:01 | **33:03** | 07:02 | 58:01 | 03:02 | 07:02 | 4 | 0.004 |
|  | ID**TTIGEWAFW**ETKK | 02:01 | 33:03 | 07:02 | **58:01** | 03:02 | 07:02 |  |  |

Bold HLA cells indicate predicted good binders (<2%) as determined by the IEDB database (IEDB.org). Bold sequence text indicates the predicted sequence which will bind to the highlighted HLA.

survivor cohort they investigated responded to EBOV GP protein whereas the majority of people responded to VP24, VP40 or NP protein[17]. Therefore, future work should look to broaden investigation into the immune response to a range of EBOV proteins with a view to incorporating greater antigenic diversity into the various vaccine platforms. The frequency of cytokine-producing T cell responses we detected when performing our ICS assays was of low magnitude, but are similar to those seen by Sakabe et al. whose cohort would have been at a similar time post EVD recovery[17].

Although highly heterogeneous, the EVD survivors we studied, in general, responded to peptide pools GP1-2 and GP1-4 which correspond to a portion of the receptor-binding domain and glycan cap of the GP. Once EBOV enters its target cell via receptor-mediated endocytosis there is progress towards a late endosome which could ultimately lead to the destruction of the virus. However, cathepsins remove a proportion of the GP including the glycan cap, this allows binding to the Niemann-Pick C1 (NPC1) and egress from the late endosome to the cell cytoplasm[25]. Since the glycan cap remains in the late endosome it could be hypothesised that further maturation results in more efficient cross-presentation of these peptides and activation of CD8+ T cells via the cytosolic or vacuolar pathways and that this is one reason why we see the majority of responses to GP1-4 which covers the glycan cap[26]. In contrast, peptide pool GP1-5 which lies in the mucin domain and covers amino acid positions 305–389 was unresponsive to the majority of survivor samples suggesting this region is not very immunogenic. The lack of activity in GP1-5 was also seen by Powlson et al. and may, be due to steric shielding, whereby the N- and O- linked glycans within the mucin domain block recognition of MHC and dampen CD8+ T cell responses[19,27].

Our flow cytometry studies show that both CD4+ and CD8+ subsets contribute to EBOV GP-specific T cell memory and that, as expected, they have different cytokine profiles, with CD4+ T cell producing IFNγ, TNFα and IL-2 whereas CD8+ T cells primarily produced only IFNγ and TNFα. Previous work in response to adenovirus vaccination in primates has suggested that an effective durable response requires double and triple cytokine producing populations, although this work was restricted to CD8+ T cells only[28]. It is likely that both CD4+ and CD8+ T cells are important for an effective immune response to EBOV as depletion of CD4+ T cells prior to rVSV vaccination and challenge in an NHP model has been shown to result in a loss of protection[12].

ELISpot and cytokine responses to individual peptides were of low magnitude and future work may wish to consider using approaches beyond cytokine detection to characterise this EBOV-specific T cell compartment[29]. Although we detected CD4+ specific responses by flow cytometry, caution should be taken when using frozen material since previous work has demonstrated that the CD4+ compartment can be compromised when using frozen PBMC[30]. However, we did find that following overnight stimulation the CD4+ phenotype of cytokine-producing cells was CCR7+/- and CD45RO+, consistent with a central memory phenotype. Whereas, CD8+ antigen-specific cells were CCR7-/+ and CD45RO-, consistent with a naïve or effector phenotype; however, these cells were also CD28+ and CD95+ (Supplementary Fig. 5) so could potentially be stem cell memory T cells[31,32]. Future work may wish to consider the use of tetramers or scRNAseq to further characterise the T cell responses that we are seeing.

## Methods

**Study designs and participants.** During 2017 a total of 62 volunteers were recruited from Guinea, Coyah. Blood was collected from 57 survivors, approximately two years post EVD. Additionally, blood was collected from negative controls, these were five EBOV naïve West African participants who had not knowingly associated or exposed themselves to EVD patients. Participants presented their EVD survivor certificate or were identified on Ebola treatment centre (ETC) databases, to verify that they were survivors. All volunteers were informed of the procedures and purpose of the study and only consenting participants were included. Ethical approval was obtained from the National Ethics Committee for Health Research, Guinea (No. 33/CNERS/15) and from the National Research Ethics Service, UK.

**ELISpot**. Freshly isolated PBMC were prepared at $2 \times 10^6$ cells/ml in supplemented Leibovitz media for IFNγ ELISpot. Cells which were frozen in liquid nitrogen were thawed using warm media and rested overnight in 5% $CO_2$ and 37 ˚C in complete RPMI media. GP peptide library at a final concentration of 2.5 μg/peptide was used to stimulate PBMC[33]. Following 18 to 20 h incubation at 37 ℃, IFNγ release was determined by standard ELISpot protocol (Mabtech 3420-2A, Sweden) and spot forming units (SFU) enumerated using an S6 core analyser (Cellular Technology Limited, Germany). IFNγ release was calculated by subtracting spots from the media only wells to determine antigen-specific counts. The results were determined as SFU's per one million cells and IFNγ responses to EBOV GP peptide were summed, to determine the overall T cell response.

**T cell depletion**. Frozen PBMC were thawed in warm media and rested overnight. The following day >$2 \times 10^6$ cells were washed and resuspended in staining buffer (PBS, 0.5% FCS, 0.5 mM EDTA). Cells were then incubated with either anti-CD4 (Miltenyi; 130-045-101) or CD8 (Miltenyi; 130-045-201) microbeads following manufacturer's instructions. Samples were then passed through a magnetised LD column and collected into 5 ml FACS tubes. Sample was stained for CD3, CD4 and CD8 and acquired on the flow cytometer to confirm depletion of the desired target cell population.

**HLA analysis**. HLA typing was performed using the Fluidigm/MiSeq NGS (next-generation sequencing) method. Briefly, ~120 ng of genomic DNA was used for PCR amplification with Fluidigm Access Array (Fluidigm Singapore PTE Ltd, Singapore). Locus-specific primers were designed to amplify a total of 23 poly-morphic exons of HLA-A, B, C (exons 1 to 4), DPA1 (exon 2), DPB1 (exons 2, 3), DQA1 (exons 1, 2, 3), DQB1 (exons 2, 3), DRB1 (exons 2, 3), and DRB3, 4, 5 (exon 2) genes. The 23 PCR amplicons were pooled, concentration adjusted, and sub-jected to sequencing on an illumina MiSeq sequencer (illumine, San Diego, CA 92122 USA). HLA alleles and genotypes were assigned using the Omixon HLA Explore (beta version) software (Omixon, Budapest, Hungary).

**Intracellular cytokine staining (ICS)**. PBMCs were resuspended in warmed complete media and rested overnight at 37 ℃ & 5% $CO_2$. The following day cells were adjusted to $2 \times 10^6$ cells/ml in media containing anti-CD28, CD49d and CD107a-PerCP cy5.5 (1 μg/ml). Samples were then left either untreated (NT) or were stimulated with EBOV GP peptide pool, containing 187 15 mer overlapping peptides at 2.5 μg/peptide or 1 μg/ml SEB for 16–18 h[2,34]. Two hours into the incubation brefeldin A and monensin (1 μg/ml) were added to block cytokine secretion from the cell. The following day samples were washed in cold FACS wash and LIVE/DEAD fixable aqua dye was added. Samples were washed, then incubated with a cell surface cocktail of antibodies including CD3-APC 750, CD4-BV786, CD8-AF700, CD19-BV510, CD14-BV510, CCR7- APC, CD95-BV395, CD45RO-BV605. Cells were then washed, fixed and permeabilised before staining for intracellular cytokines using IFNγ-AF488, TNFα- BV421 and IL-2-PE. Samples were then washed resuspended and acquired using a BD Fortessa machine and FACS Diva software (BD Biosciences, UK). Sample analysis utilised FlowJo™ v10 software[2,20].

**Reporting summary**. Further information on research design is available in the Nature Research Reporting Summary linked to this article.

## Data availability

The authors declare that the data supporting the findings of this study are available within the paper and its supplementary information files.

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

## Acknowledgements

The authors would like to acknowledge the support and commitment of Coyah and Guéckédou Ebola virus disease survivor's associations and all the participants in this study. We are also extremely grateful to the Guinean authorities and members of Centre for Training and Research on Priority Diseases including Malaria in Guinea for their support of this study. This work was funded by Food & Drug Administration, USA (Contract; HHSF223201510104C), Horizon 2020 EU, project EVIDENT (Grant Agreement No. 666100) and Wellcome-DFID grant reference 214626/Z/18/Z. This project has been funded in part with federal funds from the Frederick National Laboratory for Cancer Research, under Contract No. HHSN261200800001E. The content of this publication does not necessarily reflect the views or policies of the Department of Health and Human Services, nor does mention of trade names, commercial products, or organisations imply endorsement by the U.S. Government. This research was supported in part by the Intramural Research Programme of the NIH, Frederick National Lab, Center for Cancer Research.

## Author contributions

T.T wrote the manuscript with input from K.E., M. Carroll, M. Carrington, Y.H., S.L., J.M. and A.W. T.T, Y.H. and J.A.B were responsible for sample collection and data analysis. M. Carroll, M.K.K., S.G. and Y.H. were responsible for project conception and experimental design. A. White, L. S. Sibley, C. Sarfas aided in the design of flow cytometry studies. Y.Y., M.M. and M. Carrington performed HLA genotyping studies.

## Competing interests

The authors declare no competing interests.
