## [Peer Review File · Nature Communications]

REVIEWER COMMENTS

Reviewer #1 (Remarks to the Author):

Tipton et al describe the T cell response to the EBOV glycoprotein in 57 survivors of EVD from Guinea. This is important and relevant to the field given that the current vaccine platforms also use this antigen and the duration and magnitude of EBOV T cell responses in survivors of natural disease are not well defined, especially to GP since other viral antigens have been reported to be the major targets of virus specific T cells.

Methods and analysis as described are sound.

A few questions/points for clarification:

1. With such a precious resource, and the prior reports suggesting that NP is the major antigenic target, why weren't NP peptides also included in the study?
2. Do T cell responses correlate with GP specific antibody responses in this cohort?
3. In figure 1, it appears that GP peptide pool 1-5 is especially underrepresented? Is there anything about the structure and/or processing of GP that could explain this? Would be nice to include this in your discussion.
4. Why were only peptides 79 and 82 selected for more detailed analyses? Peptide 44 seems to have a similarly strong response in donors.
5. The discussion mention GP1-2 and GP1-4 has having the strongest responses and comments on them being in the mucin domain and the glycan ca but this does not reflect what is shown in figure 1. GP1-2 is labeled as receptor binding domain and GP1-4 is glycan cap. Please clarify.

Reviewer #2 (Remarks to the Author):

The manuscript by Tipton et al describes CD4+ and CD8+ T cell responses from EVD survivors and is only the second report to examine T cells and T cell epitopes in a large set of human survivors. Moreover, as current EBOV vaccines are focused on the GP as the sole immunogen, understanding T cell responses in survivors are critical for evaluating and comparing immunity in vaccinees. Because of this and the repeated outbreaks in the DRC, their research should be of great interest to anyone interested in ebolavirus disease.

The experiments are well thought out and the data presented in a way that is easy to evaluate the range of responses. There are several issues, however, in both the presentation of previous research and the interpretation of their results that must be addressed before publication.

Major Issues:

1. The 2014-2016 West Africa "outbreak" is considered an epidemic and should be referred to as such.
2. Line 44: The authors refer to a widely publicized study that claims 100% protection using the rVSV-ZEBOV. This result is particularly controversial (See Metzger WG, Vivas-Martínez S. Questionable efficacy of the rVSV-ZEBOV Ebola vaccine. *Lancet* 2018; 391: 1021. AND Keusch G, McAdam K, Cuff PA, Mancher M, Busta ER. Integrating clinical research into epidemic response: the Ebola experience. Washington, DC: The National Academies Press, 2017). Further evidence that this vaccine is not 100% effective is data from a recent trial in the DRC where a substantial portion of clinical EVD cases were previous vaccinees, some who had received vaccination >10 previous to the start of symptoms ("Of the 620 patients for whom information on vaccination with rVSVΔG-ZEBOV-GP was available, 155 patients (25.0%) reported that they had received the vaccine; of these, 38.7% reported that they had received the vaccine at least 10 days before the onset of clinical symptoms." from Mulango et al, *N Engl J Med* 2019). While this widely used

vaccine likely has significant efficacy, increasing evidence contradicts the initial claim of 100% efficacy.

3. In the Introduction, the authors seem to conflate protective immunity from vaccination to induced immunity during acute infection. Discussion of vaccine induced immunity and immune responses during acute infection should be separate. Further, the authors discuss the importance of T cell immunity resulting from the Ad5 EBOV vaccine but fail to mention that EBOV-specific T cell immunity is dispensable for protection from the rVSV-ZEBOV vaccine. This clearly indicates that protection, in NHPs at least, can be mediated by either T cells or B cells and seems to be dependent on the viral vector.

In relation to immune responses during acute infection, evidence for adaptive immune-mediated clearance of acute EBOV infection suggests both T cell and B cell (antibody) responses are important for survival.

4. It is unclear what the "N" value is for many of the graphs. Are all the graphs based on T cell responses from all 57 survivors? N values are given for figure 1A but not for the remaining figures.

5. There is a failure to discuss the implications of sGP in their work. This is especially important as 1) sGP is present in substantially higher levels than GP and 2) their own data and data from the Sakabe et al paper implicate the shared region between GP and sGP is the major driver of GP specific CD8+ T cell epitopes. For instance, their data show that 1-1, 1-2 and 1-3 GP peptide pools have the widest responses. These are sequences shared between GP and sGP. Only the median of 1-6 peptide pool, which is exclusive to the GP and not present in sGP, stretches just past the LLD. It is unfortunate that the authors did not also include peptide pools for the sGP. Peptide 82, present in the 1-4 pool, is the most extensively characterized epitope and is comprised of the final 15 aa acids in common between GP and sGP. As GP based vaccines do not produce sGP, this may be overlooked, but because this study is examining GP responses in survivors, this limitation should be discussed.

Also, while not discussed in Sakabe et al, peptide 82 has been entered in the IEDB database under ID: 5710297, bolstering the validity of this epitope.

Minor Issues:

1. Lines 83-84: There should be a conjunction between "pathogens" and "it". Or this sentence should be separated into two sentences.

Reviewer #3 (Remarks to the Author):

The manuscript by Tipton et al investigates Ebola virus glycoprotein specific (GP) specific CD4 and CD8 T cell responses in a cohort of Ebola Virus Disease (EVD) survivors from West Africa). Volunteers were recruited on average >1yr post infection and as such the long-term memory T cell responses were being measured in this work.

The work presented is descriptive, a well-established approach of using synthetic overlapping 15 mer peptides representing the predicted amino acid sequence to the virus GP to stimulate both CD4 and CD8 T cell responses has been used. The peptides were used in a complete GP library pool as well as smaller sub-libraries and in some experiment's individual peptides for some fine mapping work. IFN γ ELISpot assay as well as ICS Flowcytometry for IFN γ /TNF α /IL-2 in conjunction with some cell surface phenotypic analysis was also performed. Genomic HLA typing and some in-silico prediction of MHC Class I binding on a single 15 mer peptide was also presented.

Figure 1. The results show that stimulation of whole PBMC with the complete GP library elicited measurable IFN γ responses, above that of the negative control population used, the number of negative individuals used was small n=4, compared to the survivor population n=57. The analysis was repeated using smaller pools of the peptides covering GP. Positive responses were found in most pools 1-5 being the lowest. The data is highly aggregated and what is lost is any relationship of responses within an individual to the different sub-pools. A more comprehensive analysis of the available data should be undertaken.

Figure 2 Is again an ELISpot analysis which focuses in on individual peptides within GP 1-2 and GP1-4 sub-pools, as the majority of those tested responded to these pools. The results show that some donors responded to individual peptides above LLD from both pools. It is however again whole PBMC that has been stimulated in these experiments and as such it is not possible to distinguish IFN γ responses coming from either the CD4 or from CD8 T cells. The data analysis is quite simple, it is not possible to identify an individual along with which individual peptides they respond to. In addition, the individual peptides overlap each other, in some cases this may well identify fine specificity, although because it will not be clear if the response from a given individual was from a CD4 or a CD8 T cell in some instances this might be unclear. However, some of the data for responses by individuals to overlapping regions should be shown and analysis performed.

The conclusion from the work is that peptide 79 and 82 are of further interest – it is unclear what was so interesting about these particular peptides, could you please clarify?

Figure 3 Flowcytometry ICS following whole GP library stimulation with discrimination of CD4 and CD8 T cell IFN γ /TNF α /IL-2. Detection of IFN γ responses was expected in both T cell subsets, additionally IFN/TNF α dual expression (CD8) and triple IFN γ /TNF α /IL-2 (CD4) was demonstrated. This is however the complete Library and as such adds relatively little to the story, the GP sub-pools have not been used or even the GP1.2/1.4 pools or more informative the overlapping peptides.

Figure 4 Flowcytometry ICS on peptides 79 and 82. Please note line 156 references peptide 78 instead of 79. The analysis shows that 79 is recognized by CD4 T cells (IFN γ), they also make IL-2 but surprisingly DO NOT make TNF α given previous data in Fig 3 that suggested the CD4 responses tended to be triple this is a surprise and the authors should comment on the results. Peptide 82 is suggested to be contain a CD8 epitope. No LLD is shown in these graphs and it would seem that only 2 people tested had T cells specific to the peptide 82. Overlapping peptides around peptide 82 were available which could have fine mapped the response in an individual donor and might have strengthened the in-silico analysis.

Table 1 in-silico analysis identifies potential sequences that map to HLA A and B alleles within 82, however none of this has been experimentally verified even when overlapping peptides were available that could have been used.

The work presented does show that EVD survivors make CD4 and CD8 T cell responses, however the additional data do not provide for a comprehensive characterisation of CD4 and CD8 T cell responses to this protein. Additional analysis could be performed with the data available and reagents were available to extend analysis that was performed.

REVIEWER COMMENTS

Reviewer #1 (Remarks to the Author):

Tipton *et al* describe the T cell response to the EBOV glycoprotein in 57 survivors of EVD from Guinea. This is important and relevant to the field given that the current vaccine platforms also use this antigen and the duration and magnitude of EBOV T cell responses in survivors of natural disease are not well defined, especially to GP since other viral antigens have been reported to be the major targets of virus specific T cells.

Methods and analysis as described are sound.

A few questions/points for clarification:

1. With such a precious resource, and the prior reports suggesting that NP is the major antigenic target, why weren't NP peptides also included in the study?

Reply; We thank the reviewer for their reading of the manuscript and agree that NP specific results would be of interest given the heightened T cell responses reported. The primary aim of this research was to investigate the GP specific response so as to inform on the survivor compared to a vaccinated response. Another limiting factor was the volume of blood that we were able to process in the field, we collected ~30-50 ml of blood for each donor and to run ELISpot for both GP and NP would have left very little PBMC to transport back to the UK for further studies.

2. Do T cell responses correlate with GP specific antibody responses in this cohort?

Reply; We agree that comparisons to serological responses would be of interest and have compiled the following graph to highlight these correlations. We have also made mention of this in the text.

Line 113- 115; "It can also be seen in Figure 1c that the majority of EVD survivors are mounting a T cell response to the soluble region of the GP and that these T cell responses correlate with whole virus antibody levels (Supplemental figure 2)."

Supplementary figure 2: Correlations between IFN γ ELISpot and humoral immune response

Correlation between the total summed ELISpot response and matched Humoral response. A) whole virus ELISA IgG response. B) Live virus neutralisation response and C) anti-GP IgG response.

3. In figure 1, it appears that GP peptide pool 1-5 is especially underrepresented? Is there anything about the structure and/or processing of GP that could explain this? Would be nice to include this in your discussion.

Reply; GP peptide pool 1-5 contains 20 peptides (15mers) and covers amino acid positions 305 – 389 and according to PMID: 18615077 this covers the Mucin domain of the GP. Interestingly this peptide pool spans the cleavage site between full length GP and sGP which may go some way to explaining these results. Additionally, PMID: 26158395 also found no response to this region following analysis of vaccinated individuals and PMID: 30038008 suggest that most CD8 responses are to the conserved region of the GP1/sGP region. Speculatively, since this site covers the Mucin domain there may be some homology with host human mucin domains meaning that a T cell response would not be generated for fear of creating and autoreactive clone. Additionally, PMID: 20844579 demonstrated that due to the glycosylation of this region steric shielding will actively block MHC binding. We have now commented on this in the discussion.

Line 270 – 275; “In contrast, peptide pool GP1-5 which lies in the mucin domain and covers amino acid positions 305-389 was unresponsive to the majority of survivor samples suggesting this region is not very immunogenic. The lack of activity in GP1-5 was also seen by Powlson *et al* and may, be due to steric shielding, whereby the N- and O- linked glycans within the mucin domain block recognition of MHC and dampen CD8⁺ T cell responses.”^{19,27}

4. Why were only peptides 79 and 82 selected for more detailed analyses? Peptide 44 seems to have a similarly strong response in donors.

Reply; We have acquired ICS Data on peptides 36 and 44 however we could not categorically say weather responses were due to CD4 or CD8 T cells. Additionally, we acquired data on peptide 3 which is part of the SP and gave very strong response amongst a few survivors. It was found that peptide 3 gave a clear CD8 T cell response. This data has now been presented as supplemental figure 8.

Supplementary figure 8: ICS assays looking at peptide 3 and GP1-2 peptide stimulation

Intracellular cytokine staining experiments using either **A) peptide 3** (LPRDRFKRTSFFLWV) or **B) GP1-2 peptide sub pools**. Within B peptide GP1-2 sub pool 1 corresponds to peptides 29-33, GP1-2 sub pool 2 corresponds to peptides 34-40 and GP1-2 sub pool 3 corresponds to peptides 41-48.

5. The discussion mentions GP1-2 and GP1-4 has having the strongest responses and comments on them being in the mucin domain and the glycan cap but this does not reflect what is shown in figure 1. GP1-2 is labelled as receptor binding domain and GP1-4 is glycan cap. Please clarify.

Reply; We thank the reviewer for bringing this error to our attention. GP1-2 covers positions 103-180 of the EBOV GP and this does indeed relate to the receptor binding domain. GP1-4 covers positions 234-315 and this relates to the glycan cap. We apologise for any confusion caused and have updated the text accordingly;

Line 261-270; “Although highly heterogeneous, the EVD survivors we studied in general responded to peptide pools GP1-2 and GP1-4 which correspond to the **receptor binding** domain and glycan cap of the GP. Once *Ebolavirus* enters its target cell via receptor mediated endocytosis there is progress towards a late endosome which could ultimately lead to destruction of the virus. however, cathepsins remove a proportion of the GP including the glycan cap, this allows binding to the Niemann-Pick C1 (NPC1) and egress from the late endosome to the cell cytoplasm²¹. Since the glycan cap remains in the late endosome it could be hypothesised that further maturation results in more efficient cross presentation of these peptides and

activation of CD8⁺ T cells via the cytosolic or vacuolar pathways and that this is one reason why we see the majority of responses to GP1-4 which covers the glycan cap²².”

Reviewer #2 (Remarks to the Author):

The manuscript by Tipton et al describes CD4+ and CD8+ T cell responses from EVD survivors and is only the second report to examine T cells and T cell epitopes in a large set of human survivors. Moreover, as current EBOV vaccines are focused on the GP as the sole immunogen, understanding T cell responses in survivors are critical for evaluating and comparing immunity in vaccines. Because of this and the repeated outbreaks in the DRC, their research should be of great interest to anyone interested in ebolavirus disease.

The experiments are well thought out and the data presented in a way that is easy to evaluate the range of responses. There are several issues, however, in both the presentation of previous research and the interpretation of their results that must be addressed before publication.

Major Issues:

1. The 2014-2016 West Africa “outbreak” is considered an epidemic and should be referred to as such.

Reply; We thank the reviewer for their reading of our manuscript and have changed mention of outbreak to epidemic throughout.

2. Line 44: The authors refer to a widely publicized study that claims 100% protection using the rVSV-ZEBOV. This result is particularly controversial (See Metzger WG, Vivas-Martínez S. Questionable efficacy of the rVSV-ZEBOV Ebola vaccine. *Lancet* 2018; 391: 1021. AND Keusch G, McAdam K, Cuff PA, Mancher M, Busta ER. Integrating clinical research into epidemic response: the Ebola experience. Washington, DC: The National Academies Press, 2017). Further evidence that this vaccine is not 100% effective is data from a recent trial in the DRC where a substantial portion of clinical EVD cases were previous vaccines, some who had received vaccination >10 previous to the start of symptoms (“Of the 620 patients for whom information on vaccination with rVSVΔG-ZEBOV-GP was available, 155 patients (25.0%) reported that they had received the vaccine; of these, 38.7% reported that they had received the vaccine at least 10 days before the onset of clinical symptoms.” from **Mulango et al, N Engl J Med 2019**). While this widely used vaccine likely has significant efficacy, increasing evidence contradicts the initial claim of 100% efficacy.

Reply; We thank the reviewer for their comment and agree that the rVSV is likely not 100% effective and have expanded the text to highlight this important emerging data.

Line 48-53; “There are a number of large animal studies demonstrating the efficacy of the VSV-ZEBOV vaccine and a large clinical trial in Guinea has shown the vaccine to be effective and appropriate for ring vaccination strategy, with a reported 100% efficacy (95% CI 79.3–100.0; p=0.0033).⁷ However, emerging evidence suggests that there have been cases of disease breakthrough associated with this vaccine.⁸”

3. In the Introduction, the authors seem to conflate protective immunity from vaccination to induced immunity during acute infection. Discussion of vaccine induced immunity and immune responses during acute infection should be separate. Further, the authors discuss the importance of T cell immunity resulting from the Ad5 EBOV vaccine but fail to mention that EBOV-specific T cell immunity is dispensable for protection from the rVSV-ZEBOV vaccine. This clearly indicates that protection, in NHPs at least, can be mediated by either T

cells or B cells and seems to be dependent on the viral vector. In relation to immune responses during acute infection, evidence for adaptive immune-mediated clearance of acute EBOV infection suggests both T cell and B cell (antibody) responses are important for survival.

Reply; We agree that vaccine induced immunity is not comparable to naturally acquired immunity and have made efforts to distance these two things in the introduction. Additionally, we have included the study by Marzi which highlights the importance of the antibody mediated response to give a more balanced view of the subject matter.

Line 54-82; “Another vaccine candidate to be developed utilises a recombinant chimpanzee adenovirus sub group 3 virus as a vector for EBOV, Mayinga strain, GP (ChAd3-EBO-Z). Research has investigated this vaccine on its own or in combination with a modified Vaccinia Ankara (MVA-BN-Filo) boost. The MVA-BN-Filo boost encodes for the same Mayinga strain GP as the ChAd3-EBO-Z, as well as the *Sudan Ebolavirus* GP and Marburg virus GP, in addition MVA-BN-Filo encodes for the Tai-Forest *Ebolavirus* nucleoprotein (NP)^{3,9}. A close relation to this vaccine combination has been developed and has recently received marketing authorisation from the European union medicines agency¹⁰. Both candidate vaccines are continuing to show success in the field, however, to what extent these vaccines need to mediate a cellular or humoral response to provide protection is still unclear.”

Evidence from animal studies and survivor cohorts are helping us understand the naturally acquired immune response which in turn will help inform on vaccine design and may help elucidate the comparative need for a humoral or cellular response. Early work investigated the T cell response to mice vaccinated with venezuelan equine encephalitis virus replicons which expressed various *Ebolavirus* proteins. This work found murine antigen specific T cells to these *Ebolavirus* proteins, including the NP and GP. These T cells were expanded *in vitro* and adoptively transferred to *Ebolavirus* naïve mice, when mice were challenged with an adapted *Ebolavirus* strain it was found that they were protected from EVD¹⁰. Seminal evidence for the importance of T cells to EVD survival comes from the work of Sullivan *et al* who vaccinated non-human primates (NHPs) with human recombinant adenovirus serotype 5 (rAdHu5) which encoded for *Ebolavirus* GP. Cynomolgus macaques were vaccinated then exposed to Zaire *ebolavirus*. Interestingly, if post vaccinated animals underwent T cell depletion using an anti-CD3 monoclonal antibody (mAb) they lost their ability to control disease and succumbed to infection. Furthermore, if prior to challenge primates were CD8⁺ T cell depleted using a monoclonal antibody then, again, they were unable to control disease, this was not the case for CD4⁺ T cell depletion prior to challenge¹¹. However, work by Marzi *et al* investigating the role of T cells following rVSV-ZEBOV vaccination in NHPs indicated that CD8⁺ T cells were in fact dispensable and the humoral response, mediated by CD4⁺ T cells, was critical for vaccine mediated protection¹².

4. It is unclear what the “N” value is for many of the graphs. Are all the graphs based on T cell responses from all 57 survivors? N values are given for figure 1A but not for the remaining figures.

Reply; We apologise for any confusion, “N” does indicate the number of samples used and we have tried to clarify this in the figure legends.

5. There is a failure to discuss the implications of sGP in their work. This is especially important as 1) sGP is present in substantially higher levels than GP and 2) their own data and data from the Sakabe et al paper implicate the shared region between GP and sGP is the major driver of GP specific CD8+ T cell epitopes. For instance, their data show that 1-1, 1-2 and 1-3 GP peptide pools have the widest responses. These are sequences shared between GP and sGP. Only the median of 1-6 peptide pool, which is exclusive to the GP and not present in sGP, stretches just past the LLD. It is unfortunate that the authors did not also include peptide pools for the sGP. Peptide 82, present in the 1-4 pool, is the most extensively characterized epitope and is comprised of the final 15 aa acids in common between GP and sGP. As GP based vaccines do not produce sGP, this may be overlooked, but because this study is examining GP responses in survivors, this limitation should be discussed. Also, while not discussed in Sakabe et al, peptide 82 has been entered in the IEDB database under ID: 5710297, bolstering the validity of this epitope.

Reply; We acknowledge and regret that we do not present the data on sGP. We do however have this data. In addition to individual peptide pools we stimulate with two broad peptide pools named MP1 and MP2. MP1 spans sGP and consists of amino acids 1-347 where as MP2 contains peptides covering amino acids 341 – 676 which are exclusive of sGP. This data can be seen below and we have added it to figure 1.

Minor Issues:

1. Lines 83-84: There should be a conjunction between “pathogens” and “it”. Or this sentence should be separated into two sentences.

Reply: We have amended the text accordingly.

Line 103-105; IFN γ is a potent antiviral cytokine which is critical to the control and elimination of many intracellular pathogens. It is primarily produced by natural killer cells and antigen specific CD4⁺ and/or CD8⁺ T cells¹⁷

Reviewer #3 (Remarks to the Author):

The manuscript by Tipton et al investigates Ebola virus glycoprotein specific (GP) specific CD4 and CD8 T cell responses in a cohort of Ebola Virus Disease (EVD) survivors from West Africa). Volunteers were recruited on average >1yr post infection and as such the long-term memory T cell responses were being measured in this work.

The work presented is descriptive, a well-established approach of using synthetic overlapping 15 mer peptides representing the predicted amino acid sequence to the virus GP to stimulate both CD4 and CD8 T cell responses has been used. The peptides were used in a complete GP library pool as well as smaller sub-libraries and in some experiment's individual peptides for some fine mapping work. IFN γ ELISpot assay as well as ICS Flowcytometry for IFN γ /TNF α /IL-2 in conjunction with some cell surface phenotypic analysis was also performed. Genomic HLA typing and some in-silico prediction of MHC Class I binding on a single 15 mer peptide was also presented.

Figure 1. The results show that stimulation of whole PBMC with the complete GP library elicited measurable IFN γ responses, above that of the negative control population used, the number of negative individuals used was small n=4, compared to the survivor population n=57. The analysis was repeated using smaller pools of the peptides covering GP. Positive responses were found in most pools 1-5 being the lowest. The data is highly aggregated and what is lost is any relationship of responses within an individual to the different sub-pools. A more comprehensive analysis of the available data should be undertaken.

Reply part 1; We thank the reviewer for their reading of the manuscript and agree that the number of negatives used in this study was small compared to the survivor population, therefore, we have increased this number to 18 negatives. This increase is based on using negative controls that were sampled between 2016-2019 whereas previously the 4 negative controls related to those sampled in 2017 only. This new figure can be seen below (Figure 1B)

Reply part 2; Figure 1D gives an overview on the average response amongst our 57 EVD survivors to each peptide pool and is useful for drawing conclusions as to what is likely to be the most immunogenic region. Unfortunately, there is a loss in the ability to track an individual survivor's

response across peptide pools, therefore, we have added a new supplementary figure 3 which will hopefully provide this resolution, and this can be seen below.

Supplementary figure 3: Individual ELISpot response to GP peptide pools

IFN γ ELISpot using EVD survivor PBMC stimulated with EBOV GP peptide pools. The individual response by each survivor can be seen in the density plot with the darker squares representing a higher SFU/10⁶ cells.

Figure 2. Is again an ELISpot analysis which focuses in on individual peptides within GP 1-2 and GP1-4 sub-pools, as the majority of those tested responded to these pools. The results show that some donors responded to individual peptides above LLD from both pools. It is however again whole PBMC that has been stimulated in these experiments and as such it is not possible to distinguish IFN γ responses coming from either the CD4 or from CD8 T cells. The data analysis is quite simple, it is not possible to identify an individual along with which individual peptides they respond to. In addition, the individual peptides overlap each other, in some cases this may well identify fine specificity, although because it will not be clear if the response from a given individual was from a CD4 or a CD8 T cell in some instances this might be unclear. However, some of the data for responses by individuals to overlapping regions should be shown and analysis performed.

Reply Part 1; We now present additional ELISpot data showing the individual response to each peptide within our peptide pool as Figure 2A. This data was generated using PBMC from 15 survivors and was processed fresh, in the field. The individual response from each survivor can be seen as supplemental figure 4.

Reply part 2; We agree that Stimulation of Whole PBMC fractions does not inform on the CD4 or CD8 IFN γ response specifically. Therefore, we have generated more ELISpot data using CD4 and CD8 depletions to highlight the role of either the CD4 or CD8 subsets. This can now be seen below as a new Figure 3.

Figure 3: EVD survivor T cell memory response to EBOV glycoprotein GP1-4

IFN γ ELISpot using EVD survivor PBMC that were depleted of either CD4+ or CD8+ T cells A) representative flow cytometry and ELISpot images for survivor C052 showing that PBMC were successfully depleted for the desired T cell population and the corresponding ELISpot response to the various stimulation conditions. B-D) Response to GP1-4 peptide sub pools. B shows the response to sub pool 1 which consists of peptides 69-73, C shows the response to sub pool 2 which contains peptides 74-80 and D shows the response to sub pool 3 which contains peptides 81-88. Data shows the mean +SD of 11 EVD survivor samples. One-way ANOVA with repeated measures used for statistical analysis.

The conclusion from the work is that peptide 79 and 82 are of further interest – it is unclear what was so interesting about these particular peptides, could you please clarify?

Reply part 3; We have now gained additional data on other peptides of interest, namely peptide 3, 36 and 44 this is presented as supplemental figure 8. The primary reason that we perused analysis of GP1-4 and Peptides 79 and 82 is that they were previously cited as being immunogenic in either survivor or vaccinated populations.

Figure 3; Flowcytometry ICS following whole GP library stimulation with discrimination of CD4 and CD8 T cell IFN γ /TNF α /IL-2. Detection of IFN γ responses was expected in both T cell subsets, additionally IFN/TNF α dual expression (CD8) and triple IFN γ /TNF α /IL-2 (CD4) was demonstrated. This is however the complete Library and as such adds relatively little to the story, the GP sub-pools have not been used or even the GP1.2/1.4 pools or more informative the overlapping peptides.

Reply; As expected CD4 and CD8 T cells both showed activity following stimulation with all peptides, it is also interesting to see the difference and cytokine production by CD4 or CD8 T cells, something which again is perhaps expected but gives more confidence in the assay. We have added to this data set combining it with Figure 4 to give a more focused view on the T cell response.

Figure 4 ; Flowcytometry ICS on peptides 79 and 82. Please note line 156 references peptide 78 instead of 79. The analysis shows that 79 is recognized by CD4 T cells (IFN γ), they also make IL-2 but surprisingly DO NOT make TNF α given previous data in Fig 3 that suggested the CD4 responses tended to be triple this is a surprise and the authors should comment on the results. Peptide 82 is suggested to be contain a CD8 epitope. No LLD is shown in these graphs and it would seem that only 2 people tested had T cells specific to the peptide 82. Overlapping peptides around peptide 82 were available which could have fine mapped the response in an individual donor and might have strengthened the in-silico analysis.

Reply part 1; We thank the reviewer for bringing this error to our attention and have corrected reference of peptide 78 to 79. With regards to CD4 T cell response to peptide 79 we found that CD4 T cells produced all three cytokines (IFN, TNF and IL2), Figure 3 uses Boolean gating to show that activated CD4 T cells are producing all three cytokines and as opposed to CD8 T cells are not producing IFN+ TNF α in the absence of IL-2 we apologise for any confusion and have tried to make this clearer in the text.

Reply part 2: Two survivor do show a good CD8 specific response to peptide 82, in hindsight it was perhaps ambitious to stimulate with one 15mer peptide and results would have been clearer if we had used a number of surrounding peptides but due to a lack of PBMC we would be unable to repeat

these stimulations. We have, however, collected more ICS data using surrounding peptide for peptides 36 and 44 which can now be seen in supplemental figure 8.

Reply part 3; we apologise and have now quantified an LLD based on the data available, this can now be seen as part of figure 4.

Table 1; in-silico analysis identifies potential sequences that map to HLA A and B alleles within 82, however none of this has been experimentally verified even when overlapping peptides were available that could have been used.

Reply; Based on the cellular data generated we predict using in silico methods which HLA are likely to bind peptide 82. To experimentally verify these finding would need to run MHCI binding assays using cells that express a single MHCI molecule. It would be of interest to experimentally verify these in silico predictions however, we feel this would require a significant amount of further research which is beyond our expertise, additionally, the focus of this study which was primarily to highlight T cell peptides amongst Survivor PBMC and to compare with what is known about vaccine induced responses.

The work presented does show that EVD survivors make CD4 and CD8 T cell responses, however the additional data do not provide for a comprehensive characterisation of CD4 and CD8 T cell responses to this protein. Additional analysis could be performed with the data available and reagents were available to extend analysis that was performed.

Reply: We again thank the reviewer for their reading of the manuscript and hope that the above replies will satisfy all comments. In particular the addition of Figure 2A which show very fine mapping of the IFN γ response to each peptide amongst 15 survivors and Figure 3 which used T cell specific depletions to demonstrated either a CD4 or CD8 response.

REVIEWERS' COMMENTS

Reviewer #1 (Remarks to the Author):

My comments have been addressed and the manuscript is improved.

Reviewer #2 (Remarks to the Author):

The authors have done a superb job addressing comments from the reviewers. I have no further comments or criticisms of this manuscript.

Brian M. Sullivan

Reviewer #3 (Remarks to the Author):

I thank the authors for their comprehensive answers to the comments I had made in my review. In addition I acknowledge that the authors have made a considerable effort to address the comments with additional experimentation and analysis.

I think they have done a very good job of addressing the points.